# Maximal Divergence Sequential Auto-encoder for Binary Software Vulnerability Detection*

**Tue Le** [†]**, Tuan Nguyen** [†]
AI Research Lab, Trusting Social, Australia
`{tue.le, tuan.nguyen}@trustingsocial.com`

**Trung Le, Dinh Phung**
Monash University, Australia
`{trunglm, dinh.phung}@monash.edu`

**Paul Montague, Olivier De Vel**
Defence Science and Technology Group, Department of Defence, Australia
`{paul.montague, olivier.devel}@dst.defence.gov.au`

**Lizhen Qu**
Data61, CSIRO, Australia
`lizhen.qu@data61.csiro.au`

## ABSTRACT

Due to the sharp increase in the severity of the threat imposed by software vulnerabilities, the detection of vulnerabilities in binary code has become an important concern in the software industry, such as the embedded systems industry, and in the field of computer security. However, most of the works in binary code vulnerability detection has relied on handcrafted features which are manually chosen by a select few domain experts. In this paper, we attempt to alleviate this severe binary vulnerability detection bottleneck by leveraging recent advances in deep learning representations and propose the Maximal Divergence Sequential Auto-Encoder. In particular, latent codes representing vulnerable and non-vulnerable binaries are encouraged to be maximally divergent, while still being able to maintain crucial information from the original binaries. We conducted extensive experiments to compare and contrast our proposed methods with the baselines, and the results indicate that our proposed methods outperform the baselines in all performance measures of interest.

## 1 INTRODUCTION

Software vulnerabilities are specific flaws or oversights in a piece of software that allow attackers to perform malicious acts including exposing or altering sensitive information, disrupting or destroying a system, or taking control of a computer system or program (Dowd et al., 2006). Due to the ubiquity of computer software, the growth and the diversity in its development process, a great amount of computer software potentially includes software vulnerabilities. This fact makes the problem of software security vulnerability identification an important concern in the software industry and in the field of computer security. Although a great effort has been made by the security community, the severity of the threat from software vulnerabilities has gradually increased over the years. Numerous

---

*__Acknowledgement__: This research was partially supported under the Defence Science and Technology Group's Next Generation Technologies Program. Dinh Phung and Trung Le further acknowledge the partial support from the Australian Research Council DP160109394.

[†]This work was done during the time when the first and second authors carried out internship at Monash University, Australia.

exist of examples and incidents in the past two decades in which software vulnerabilities have imposed significant damages to companies and individuals (Ghaffarian & Shahriari, 2017). For example, vulnerabilities in popular browser plugins have threatened the security and privacy of millions of Internet users (e.g., Adobe Flash Player (US-CERT 2015; Adobe Security Bulletin 2015) and Oracle Java (US-CERT 2013)), vulnerabilities in popular and fundamental open-source software have also threatened the security of thousands of companies and their customers around the globe (e.g., Heartbleed (Codenomicon 2014) and ShellShock (Symantec Security Response 2014).

Software vulnerability detection (SVD) can be categorized into source code and binary code vulnerability detection. Source code vulnerability detection has been widely studied in a variety of works (Shin et al., 2011; Neuhaus et al., 2007; Yamaguchi et al., 2011; Li et al., 2016; Kim et al., 2017; Li et al., 2018). Most of the previous work in source code vulnerability detection (Neuhaus et al., 2007; Shin et al., 2011; Yamaguchi et al., 2011; Li et al., 2016; Kim et al., 2017) has been based on handcrafted features which are manually chosen by a limited number of domain experts. To mitigate the dependency on handcrafted features, the use of automatic features in SVD has been studied recently in (Dam et al., 2017; Li et al., 2018; Lin et al., 2018). In particular, Dam et al. (2017); Lin et al. (2018) employed a Recurrent Neural Network (RNN) to transform sequences of code tokens to vectorial features, which are further fed to a separate classifier, while Li et al. (2018) combined learning the vector representation and the training of the classifier in a deep network.

Compared with source code vulnerability detection, binary code vulnerability detection is significantly more difficult because much of the syntactic and semantic information provided by high-level programming languages is lost during the compilation process. The existence of such syntactic and semantic information makes it easier to reason how data and inputs drive the paths of execution. Unfortunately, a software binary, such as proprietary binary code (with no access to source code) or embedded systems code, is generally all that is made available for code analysis (together perhaps with the processor architecture such as x86 etc.). The ability to detect the presence or absence of vulnerabilities in binary code, without getting access to source code, is therefore a major importance in the context of computer security. Some work has been proposed to detect vulnerabilities at the binary code level when source code is not available, notably work based on fuzzing, symbolic execution (Cadar & Sen, 2013; Avancini & Ceccato, 2013; Meng et al., 2016), or techniques using handcrafted features extracted from dynamic analysis (Grieco et al., 2016; Cozzie et al., 2008; White & Lüttgen, 2013). To the best of our knowledge, there has been no work studying the use of automatically extracted features for binary code vulnerability detection, though there has been some work using automatic features in conjunction with deep learning methods for malware detection, notably (Saxe & Berlin, 2015; Raff et al., 2017). It is worth noting that binary code vulnerability detection and malware detection are two different tasks. In particular, binary code vulnerability detection aims to detect specific flaws or oversights in binary code, while malware detection aims to detect if a given binary is malicious or not. The former is arguably harder in the sense that vulnerable and non-vulnerable binaries might be only slightly different, while there might be a clearer difference in general between malware and benign binaries.

In addition, a significant constraint in research on binary code vulnerability detection is the lack of suitable binaries labeled as either vulnerable or non-vulnerable. Although we have some source code datasets for software vulnerability detection, to the best of our knowledge, there exists no large public binary dataset for the purpose of binary code vulnerability detection. The reason is that most source code in source code vulnerability detection datasets is not compilable due to incompleteness, and they have important pieces missing (e.g., variables, data types) and relevant libraries – making the code compilable take a large effort in fixing a vast volume of source code. This arises from the nature of the process that involves collecting and labeling source code wherein we start from security reports in CVE[1] and navigate through relevant websites to obtain code snippets of vulnerable and non-vulnerable source code.

In this work, we leverage recent advances in deep learning to derive the automatic features of binary code for vulnerability detection. In particular, we view a given binary as a sequence of machine instructions and then use the theory of Variational Auto-Encoders (VAE) (Kingma & Welling, 2013) to develop the *Maximal Divergence Sequential Auto-Encoder* (MDSAE) that can work out representations of binary code in such a way that representations of vulnerable and non-vulnerable binaries are encouraged to be maximally different for vulnerability detection purposes,

---

[1]https://cve.mitre.org/

while still preserving crucial information inherent in the original binaries. In contrast to the original VAE wherein the data prior is kept fixed, we propose using two learnable Gaussian priors, one for each class. Based on the VAE principle, latent codes (i.e., data representations) are absorbed (or compressed) into the data prior distribution, we further propose maximizing a divergence (e.g., Wasserstein (WS) distance or Kullback-Leibler (KL) divergence) between these two priors to separate representations of vulnerable and non-vulnerable binaries. Our MDSAE can be used to produce data representations for another independent classifier (e.g., Support Vector Machine or Random Forest) or incorporated with a shallow feedforward neural network built on top of the latent codes for simultaneously training both the mechanism to generate data representations and the classifier. The former is named MDSAE-R and the latter is named MDSAE-C. We summarize our contributions in this paper as follows:

- We propose a novel method named *Maximal Divergence Sequential Auto-Encoder* (MDSAE) that leverages recent advances in deep learning representation (namely, VAE) for binary code vulnerability detection.

- One of our most significant contributions is to create a labeled dataset for use in binary code vulnerability detection. In particular, we used the source code in the published NDSS18 dataset used in (Li et al., 2018) and then extracted vulnerable and non-vulnerable functions. We developed a tool that can automatically detect the syntactical errors in a given piece of source code, fix them, and finally compile the fixed source code into binaries for various platforms (both Windows OS and Linux OS) and architectures (both x86 and x86-64 processors). Specifically, after preprocessing and filtering out identical functions from the NDSS18 *source code* dataset, we obtain $13,000$ functions of which $9,000$ are able to be fixed and compiled to binaries. By compiling the source code of these functions under the various platform and architecture options, we obtain $32,281$ binary functions including $17,977$ binaries for Windows and $14,304$ binaries for Linux.

- We conducted extensive experiments on the NDSS18 *binary* dataset. The experimental results show that the two variants MDSAE-R and MDSAE-C outperform the baselines in all performance measures of interest. It is not surprising that MDSAE-C achieves higher predictive performances compared with MDSAE-R, but the fact that MDSAE-R achieves good predictive performances confirms our hypothesis of encouraging the separation in representations of data in different classes so that a simple linear classifier subsequently trained on these data representations can obtain good predictive results.

## 2 RELATED BACKGROUND

### 2.1 THE VARIATIONAL AUTO-ENCODER

The Variational Auto-Encoder (VAE) (Kingma & Welling, 2013) is a probabilistic auto-encoder that takes into account both the reconstruction of true samples and generalization of samples generated from a latent space. The underlying idea is to learn a probabilistic decoder $p_\theta(\mathbf{x} \mid \mathbf{z})$, $\mathbf{z} \sim \mathcal{N}(\mathbf{0}, \mathbb{I})$ that can mimic the true data sample $\mathbf{x}^1, \ldots, \mathbf{x}^N$ drawn from an existing but unknown data distribution $p_d(\mathbf{x})$. VAE is developed based on the following lower bound:

$$\log p_\theta(\mathbf{x}) \geq \mathcal{L}(\mathbf{x}; \theta, \phi) = \mathbb{E}_{q_\phi(\mathbf{z}|\mathbf{x})}[\log p_\theta(\mathbf{x} \mid \mathbf{z})] - D_{KL}(q_\phi(\mathbf{z} \mid \mathbf{x}) \| p(\mathbf{z}))$$

where $q_\phi(\mathbf{z} \mid \mathbf{x})$ is the approximate posterior distribution.

We need to maximize the log likelihood at each training example $\mathbf{x}$. Therefore the objective function is of the following form:

$$\max_{\theta, \phi} \mathbb{E}_{\mathbf{x}} \left[ \mathbb{E}_{q_\phi(\mathbf{z}|\mathbf{x})}[\log p_\theta(\mathbf{x} \mid \mathbf{z})] - D_{KL}(q_\phi(\mathbf{z} \mid \mathbf{x}) \| p(\mathbf{z})) \right] \tag{1}$$

where $\mathbf{x}$ is drawn from the empirical data distribution.

To reduce the variance when using Monte Carlo (MC) estimation for tackling the above optimization problem, the reparameterization trick is employed. More specifically, assuming that $q_\phi(\mathbf{z} \mid \mathbf{x}) = \mathcal{N}(\mathbf{z} \mid \mu_\phi(\mathbf{x}), \text{diag}(\sigma_\phi(\mathbf{x})))$, we can do reparameterization as: $\mathbf{z} = \mu_\phi(\mathbf{x}) + \text{diag}(\sigma_\phi(\mathbf{x}))^{1/2} \boldsymbol{\epsilon}$ where the source of randomness $\boldsymbol{\epsilon} \sim \mathcal{N}(\mathbf{0}, \mathbb{I})$ and $\mu_\phi(\mathbf{z}), \sigma_\phi(\mathbf{z})$ are two neural networks representing the mean and covariance matrix of the approximate Gaussian posterior.

The optimization problem in Eq. (1) can be equivalently rewritten as:

$$\max_{\theta,\phi} \mathbb{E}_{\mathbf{x}} \left[ \mathbb{E}_{\boldsymbol{\epsilon}} \left[ \log p_\theta \left( \mathbf{x} \mid \mu_\phi(\mathbf{x}) + \mathrm{diag}\left(\sigma_\phi(\mathbf{x})\right)^{1/2} \boldsymbol{\epsilon} \right) \right] - D_{KL} \left( q_\phi(\mathbf{z} \mid \mathbf{x}) \| p(\mathbf{z}) \right) \right] \quad (2)$$

The first term in Eq. (2) is regarded as the reconstruction term and the second term in this equation is regarded as the regularization term. In this term, we minimize $\mathbb{E}_{\mathbf{x}} \left[ D_{KL} \left( q_\phi(\mathbf{z} \mid \mathbf{x}) \| p(\mathbf{z}) \right) \right]$, hence trying to compress and squash the latent codes $\mathbf{z}$ for each true example $\mathbf{x}$ into those sampled from the prior distribution $p(\mathbf{z})$. This observation is the key ingredient for us to develop our proposed model.

## 2.2 The Kullback-Leibler Divergence and L2 Wasserstein Distance

Given two distributions with the probability density functions $p(\mathbf{z})$ and $q(\mathbf{z})$ where $\mathbf{z} \in \mathbb{R}^d$, the Kullback-Leibler (KL) divergence between these two distributions is defined as:

$$D_{KL}(q\|p) = \int q(\mathbf{z}) \log \frac{q(\mathbf{z})}{p(\mathbf{z})} d\mathbf{z}$$

Another divergence of interest is L2 Wasserstein (WS) distance with the cost function $c(\mathbf{z}_1, \mathbf{z}_2) = \|\mathbf{z}_1 - \mathbf{z}_2\|_2^2$. The L2 Wasserstein divergence between two distributions is defined as:

$$D_{WS}(q\|p) = \min_{\pi \in \Pi(q,p)} \mathbb{E}_{(\mathbf{z}_1, \mathbf{z}_2) \sim \pi} \left[ \|\mathbf{z}_1 - \mathbf{z}_2\|_2^2 \right]$$

where $\Pi(q, p)$ specifies the set of all joint distributions over $p, q$ which admits $p, q$ as marginals.

If $p, q$ are two Gaussians, i.e., $p(\mathbf{z}) = \mathcal{N}(\mathbf{z} \mid \mu_1, \Sigma_1)$ and $q(\mathbf{z}) = \mathcal{N}(\mathbf{z} \mid \mu_2, \Sigma_2)$ then both KL divergence and L2 WS distance can be computed in close forms as:

$$D_{KL}(q\|p) = \frac{1}{2} \left[ \log \frac{|\Sigma_1|}{|\Sigma_2|} - d\mathrm{tr}\left(\Sigma_1^{-1}\Sigma_2\right) + (\mu_1 - \mu_2)^\mathsf{T} \Sigma_1^{-1} (\mu_1 - \mu_2) \right]$$

$$D_{WS}(q\|p) = \|\mu_1 - \mu_2\|_2^2 + \left\| \Sigma_1^{1/2} - \Sigma_2^{1/2} \right\|_F^2$$

where $\|\cdot\|_F$ is the Frobenius norm and $\Sigma_1 \Sigma_2 = \Sigma_2 \Sigma_1$.

## 3 The Maximal Divergence Sequential Auto-encoder (MDSAE) for Binary Vulnerability Detection

### 3.1 Data Processing and Embedding

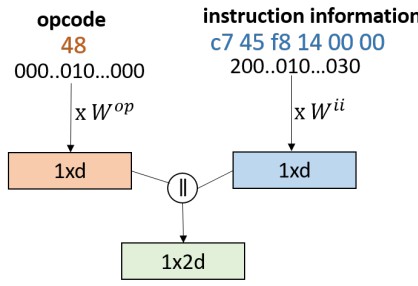

Figure 1: Machine instruction embedding.

For each machine instruction, we employ the Capstone[2] binary disassembly framework to detect entire machine instructions. We then eliminate redundant prefixes to obtain the core parts that contain the opcode and other significant information. Each core part in a machine instruction consists of two parts: the opcode and instruction information (i.e., memory location, registers, etc.). We embed both the opcode and instruction information into vectors and then concatenate them. To embed the opcode, we build a vocabulary of opcodes and then multiply the one-hot vector of the opcode with the corresponding embedding matrix. To embed the instruction information, we build the vocabulary over 256 hex-bytes from 00 to $FF$, then view the instruction information as a sequence of hex-bytes to construct the frequency vector of a size 256, and finally multiply this frequency vector with the corresponding embedding matrix. More specifically, the output embedding is $\mathbf{e} = \mathbf{e}_{op} \parallel \mathbf{e}_{ii}$ where $\mathbf{e}_{op} = \text{one-hot}(op) \times W^{op}$ and $\mathbf{e}_{ii} = \text{freq}(ii) \times W^{ii}$ with the opcode $op$, the instruction information $ii$ and its frequency vector $\text{freq}(ii)$, and the embedding matrices $W^{op}$ and $W^{ii}$. The process of embedding machine instructions is presented in Figure 1.

[2] www.capstone-engine.org

### 3.2 PROPOSED MODEL

In this work, we view binary code $\mathbf{x}$ as a sequence of machine instructions, i.e., $\mathbf{x} = [\mathbf{x}_i]_{i=1,\dots,m}$ where each $\mathbf{x}_i$ is a machine instruction. Our idea is to encode $\mathbf{x}$ to the latent code $\mathbf{z}$ in such a way that the latent codes of data in different classes are encouraged to be maximally divergent. Let us denote the distributions of vulnerable and non-vulnerable sequences by $p^1(\mathbf{x})$ and $p^0(\mathbf{x})$ respectively. Inspired by the Variational Auto-Encoder (Kingma & Welling, 2013), we propose to use a probabilistic decoder $p_\theta(\mathbf{x} \mid \mathbf{z})$ such that for $\mathbf{z} \sim p^0(\mathbf{z})$, $\mathbf{x}$ drawn from $p_\theta(\mathbf{x} \mid \mathbf{z})$ can mimic those drawn from $p^0(\mathbf{x})$ and for $\mathbf{z} \sim p^1(\mathbf{z})$, $\mathbf{x}$ drawn from $p_\theta(\mathbf{x} \mid \mathbf{z})$ can mimic those drawn from $p^1(\mathbf{x})$. In other words, we aim to learn the probabilistic decoder $p_\theta(\mathbf{x} \mid \mathbf{z})$ satisfying:

$$p^0(\mathbf{x}) = \int p_\theta(\mathbf{x} \mid \mathbf{z}) p^0(\mathbf{z}) d\mathbf{z} \quad \text{and} \quad p^1(\mathbf{x}) = \int p_\theta(\mathbf{x} \mid \mathbf{z}) p^1(\mathbf{z}) d\mathbf{z}$$

For any approximate posterior $q_\phi(\mathbf{z} \mid \mathbf{x})$, we have the following lower bounds:

$$\log p^k(\mathbf{x}) \geq \mathcal{L}^k(\mathbf{x}; \theta, \phi) = \mathbb{E}_{q_\phi(\mathbf{z}|\mathbf{x})}[\log p_\theta(\mathbf{x} \mid \mathbf{z})] - D_{KL}(q_\phi(\mathbf{z} \mid \mathbf{x}) \| p^k(\mathbf{z})), \ k = 0, 1$$

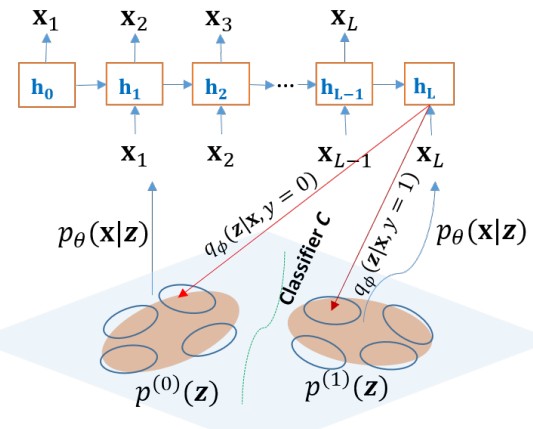

Figure 2: Maximal divergence sequential auto-encoder. The latent codes of vulnerable and non-vulnerable are encouraged to be maximally divergent, while still maintaining crucial information from the original binaries. Note that we use the same network for $q_\phi(\mathbf{z} \mid \mathbf{x}, y = 0)$ and $q_\phi(\mathbf{z} \mid \mathbf{x}, y = 1)$ and they are discriminated by the source of data used to fit.

Using the architecture shown in Figure 2, we consider the probabilistic decoder $p_\theta(\mathbf{x} \mid \mathbf{z})$ of the following parametric form

$$p_\theta(\mathbf{x} \mid \mathbf{z}) = \prod_{i=1}^{L} p_\theta(\mathbf{x}_i \mid \mathbf{x}_{1:i-1}, \mathbf{z}) = \prod_{i=1}^{L} p_\theta(\mathbf{x}_i \mid \mathbf{h}_{i-1}, \mathbf{z})$$

and hence we can further derive the lower bounds as:

$$\mathcal{L}^k(\mathbf{x}; \theta, \phi) = \mathbb{E}_{q_\phi(\mathbf{z}|\mathbf{x})}\left[\sum_{i=1}^{L} \log p_\theta(\mathbf{x}_i \mid \mathbf{h}_{i-1}, \mathbf{z})\right] - D_{KL}(q_\phi(\mathbf{z} \mid \mathbf{h}_L) \| p^k(\mathbf{z})), \ k = 0, 1$$

We arrive at the following optimization problem:

$$\max_{\theta, \phi} \left\{ \mathbb{E}_{\mathbf{x}:y=0}\left[\mathcal{L}^0(\mathbf{x}; \theta, \phi)\right] + \mathbb{E}_{\mathbf{x}:y=1}\left[\mathcal{L}^1(\mathbf{x}; \theta, \phi)\right] \right\}$$

It is worth noting that since we are minimizing:

$$\mathbb{E}_{\mathbf{x}:y=0}\left[D_{KL}(q_\phi(\mathbf{z} \mid \mathbf{h}_L) \| p^0(\mathbf{z}))\right] + \mathbb{E}_{\mathbf{x}:y=1}\left[D_{KL}(q_\phi(\mathbf{z} \mid \mathbf{h}_L) \| p^1(\mathbf{z}))\right]$$

the encoding $\mathbf{z} \sim q_\phi(\mathbf{z} \mid \mathbf{h}_L)$ with $y = 0$ are absorbed (compressed) into the prior $p^0(\mathbf{z})$. Similarly, the encoding $\mathbf{z} \sim q_\phi(\mathbf{z} \mid \mathbf{h}_L)$ with $y = 1$ are compressed into the prior $p^1(\mathbf{z})$. Therefore, to maximize the difference between the encodings of data in the two classes, we propose to maximize the divergence between $p^0(\mathbf{z})$ and $p^1(\mathbf{z})$ and arrive the following optimization problem:

$$\max_{\theta, \phi} \left\{ \mathbb{E}_{\mathbf{x}:y=0}\left[\mathcal{L}^0(\mathbf{x}; \theta, \phi)\right] + \mathbb{E}_{\mathbf{x}:y=1}\left[\mathcal{L}^1(\mathbf{x}; \theta, \phi)\right] + \alpha D(p^0(\mathbf{z}) \| p^1(\mathbf{z})) \right\}$$

where $\alpha > 0$ is a non-negative trade-off parameter and $D\left(p^0\left(\mathbf{z}\right)\|p^1\left(\mathbf{z}\right)\right)$ is the divergence between the two priors.

To facilitate the evaluation, we endow these two priors with Gaussian distributions as follows: $p^k\left(\mathbf{z}\right) = \mathcal{N}\left(\mathbf{z} \mid \boldsymbol{\mu}_k, \Sigma_k\right)$, $k = 0, 1$. We also propose using the Gaussian approximate posterior as: $q_\phi\left(\mathbf{z} \mid \mathbf{h}_L\right) = \mathcal{N}\left(\mathbf{z} \mid \mu_\phi\left(\mathbf{h}_L\right), \operatorname{diag}\left(\sigma_\phi\left(\mathbf{h}_L\right)\right)\right)$ which enables the reparameterization trick: $\mathbf{z} = \mu_\phi\left(\mathbf{h}_L\right) + \operatorname{diag}\left(\sigma_\phi\left(\mathbf{h}_L\right)\right)^{1/2}\boldsymbol{\epsilon}$, where the source of randomness $\boldsymbol{\epsilon} \sim \mathcal{N}\left(\mathbf{0}, \mathbb{I}\right)$ and $\mu_\phi\left(\mathbf{z}\right)$, $\sigma_\phi\left(\mathbf{z}\right)$ are two neural networks representing the mean and covariance matrix of the approximate Gaussian posterior.

Hence we come to the following optimization problem:

$$\max_{\theta,\phi,\boldsymbol{\mu}_0,\Sigma_0,\boldsymbol{\mu}_1,\Sigma_1} \left\{ \mathbb{E}_{\mathbf{x}:y=0}\left[\mathcal{L}^0\left(\mathbf{x};\theta,\phi\right)\right] + \mathbb{E}_{\mathbf{x}:y=1}\left[\mathcal{L}^1\left(\mathbf{x};\theta,\phi\right)\right] + \alpha D\left(p^0\left(\mathbf{z}\right)\|p^1\left(\mathbf{z}\right)\right) \right\} \quad (3)$$

where we note that $D\left(p^0\left(\mathbf{z}\right)\|p^1\left(\mathbf{z}\right)\right)$ is tractable for both the KL-divergence and L2 Wasserstein distance (See Section 2.2) and $\mathcal{L}^0\left(\mathbf{x};\theta,\phi\right)$, $\mathcal{L}^1\left(\mathbf{x};\theta,\phi\right)$ can be rewritten using the reparameterization trick as:

$$\mathcal{L}^k\left(\mathbf{x};\theta,\phi\right) = \mathbb{E}_{\boldsymbol{\epsilon}\sim\mathcal{N}(\mathbf{0},\mathbb{I})}\left[\sum_{i=1}^{L}\log p_\theta\left(\mathbf{x}_i \mid \mathbf{h}_{i-1}, \mathbf{z}\right)\right] - D_{KL}\left(q_\phi\left(\mathbf{z} \mid \mathbf{h}_L\right)\|p^k\left(\mathbf{z}\right)\right), \ k = 0, 1$$

with $\mathbf{z} = \mu_\phi\left(\mathbf{h}_L\right) + \operatorname{diag}\left(\sigma_\phi\left(\mathbf{h}_L\right)\right)^{1/2}\boldsymbol{\epsilon}$.

To classify data, we can train a classifier $\mathcal{C}$ over the latent space either independently or simultaneously with the maximal divergence auto-encoder. If we train the classifier simultaneously, the final optimization problem is as follows:

$$\max_{\theta,\phi,\psi,\boldsymbol{\mu}_0,\Sigma_0,\boldsymbol{\mu}_1,\Sigma_1} \left\{ f\left(\theta,\phi,\psi,\boldsymbol{\mu}_0,\Sigma_0,\boldsymbol{\mu}_1,\Sigma_1\right) \right\}$$

where $f\left(\theta,\phi,\psi,\boldsymbol{\mu}_0,\Sigma_0,\boldsymbol{\mu}_1,\Sigma_1\right) = \mathbb{E}_{\mathbf{x}:y=0}\left[\mathcal{L}^0\left(\mathbf{x};\theta,\phi\right)\right] + \mathbb{E}_{\mathbf{x}:y=1}\left[\mathcal{L}^1\left(\mathbf{x};\theta,\phi\right)\right] + \alpha D\left(p^0\left(\mathbf{z}\right)\|p^1\left(\mathbf{z}\right)\right) + \beta\left(\mathbb{E}_{\mathbf{x}:y=0}\left[\log\left(1 - \mathcal{C}_\psi\left(\mathbf{x}\right)\right)\right] + \mathbb{E}_{\mathbf{x}:y=1}\left[\log \mathcal{C}_\psi\left(\mathbf{x}\right)\right]\right)$ where $\mathcal{C}_\psi\left(\mathbf{x}\right)$ stands for the probability to classify $\mathbf{x}$ as a vulnerable binary code ($y = 1$), and $\alpha, \beta > 0$ are two non-negative trade-off parameters.

It is worth noting that to model the conditional distributions $p_\theta\left(\mathbf{x}_i \mid \mathbf{h}_{i-1}, \mathbf{z}\right)$, we only take into account the opcode of the machine instruction $\mathbf{x}_i$. Since this opcode lies in a fixed vocabulary of the opcodes, we can use the softmax distribution to define the corresponding distribution $p_\theta\left(\mathbf{x}_i \mid \mathbf{h}_{i-1}, \mathbf{z}\right)$. By this means, the reconstruction phase aims to reconstruct the opcodes of the machine instructions in a given binary rather than the whole machine instructions.

## 4 EXPERIMENTS

### 4.1 EXPERIMENTAL DATASETS

One of the most significant contributions of our work is to create a labeled binary dataset for binary code vulnerability detection. We first extracted the functions from the NDSS18 *source code* dataset. We then preprocessed and filtered out any identical functions to obtain $13,000$ functions, of which $9,000$ could be fixed to compile to binaries using our automatic tool. In addition, we developed a tool based on Joern[3] to parse the semantic and syntactical relationships in a given piece of source code. In particular, our tool first used the compiler gcc/g++ (MinGW) to compile a given piece of source code, then captured the error messages, parsed these error messages, relied on Joern to be aware of the semantic and syntactical relationships of the error messages with respect to the source code, and finally fixed the corresponding error message. This process was repeated until the given source is error-free and ready to compile to a binary. By compiling the compilable function source code under various platforms and architectures, we obtained $32,281$ binary functions including $17,977$ binaries for Windows and $14,304$ binaries for Linux. The statistics of our binary dataset is given in Table 1. Additionally, in order to obtain this binary dataset our tool fixed tens of thousands of errors of which many are strongly associated with specific source code.

### 4.2 BASELINES

---

[3]http://mlsec.org/joern/

|  | #Non-vulnerable | #Vulnerable | #Binaries |
|---|---|---|---|
| **Windows** | $8,999$ | $8,978$ | $17,977$ |
| **Linux** | $6,955$ | $7,349$ | $14,304$ |
| **Whole** | $15,954$ | $16,327$ | $32,281$ |

Table 1: The statistics of our binary funtions dataset.

We compared our proposed methods MDSAE-R (for learning maximally divergent representations in conjunction with an independent linear classifier to classify vulnerable and non-vulnerable functions) and MDSAE-C (for learning maximally divergent representations incorporating a linear classifier) with the following baselines:

- **RNN-R**: A Recurrent Neural Network (RNN) for learning representations and linear classifier independently trained on the resulting representations for classifying vulnerable and non-vulnerable functions. In addition, to learn representations in an unsupervised manner, we applied the method of language modeling whereby we trained the model to predict the opcode of the next machine instruction given the previous machine instructions.

- **RNN-C**: A RNN with a linear classifier built on the top of the last hidden unit.

- **Para2Vec**: The paragraph-to-vector distributional similarity model proposed in (Le & Mikolov, 2014). This work proposed to embed paragraphs including many words in a fixed vocabulary into a vector space. To apply this work in our context, we view a binary as a sequence of opcodes residing in the fixed vocabulary of the opcodes.

- **SeqVAE-C**: Sequential VAE as in Section 3.2, but we set two priors to $\mathcal{N}(\mathbf{0}, \mathbb{I})$ and kept fixed during training as in the original VAE. A linear classifier was built up on the top of the latent codes and trained simultaneously. With this setting, we aim to show that learning the priors produces more separable representations, hence boosts the performance.

- **VulDeePecker**: proposed in (Li et al., 2018) for source code vulnerability detection. This model employed a Bidirectional RNN (BRNN) to take sequential inputs and then concatenated hidden units to input to a feedforward neural net classifier. This method can inherently be applied to binaries wherein sequences of machine instructions are inputted to the BRNN.

In addition, we also inspected two variants of divergence (i.e., KL divergence and L2 WS distance (See Section 2.2)) for formulating the divergence $D\left(p^0(\mathbf{z}) \| p^1(\mathbf{z})\right)$ in the optimization problem in Eq. (3). Consequently, we have four variants of our proposed method, namely MDSAE-RKL, MDSAE-RWS, MDSAE-CKL, and MDSAE-CWS.

## 4.3 PARAMETER SETTING

We split the data into 80% for training, 10% for validation, and the remaining 10% for testing. We employed a dynamic RNN to tackle the variation in the number of machine instructions of the functions. For the RNN baselines and our models, the size of hidden unit was set to 256. For our model, the size of the latent space was set to 4,096, the trade-off parameters $\alpha$, $\beta$ were set to $2 \times 10^{-2}$ and $10^{-4}$ respectively. We used the Adam optimizer (Kingma & Ba, 2014) with an initial learning rate equal to 0.0001. The minibatch size was set to 64 and the number of epochs was set to 100. We implemented our proposed method in Python using Tensorflow (Abadi et al., 2016), an open-source software library for Machine Intelligence developed by the Google Brain Team. The source code, as well as the dataset, is available in our GitHub repository[4]. We ran our experiments on a computer with an Intel Xeon Processor E5-1660 which had 8 cores at 3.0 GHz and 128 GB of RAM.

## 4.4 EXPERIMENTAL RESULTS

### 4.4.1 EXPERIMENTAL RESULTS ON THE NDSS18 BINARY DATASET

We conducted the experiments on the subset of Windows binaries, the subset of Linux binaries, and the whole set of binaries to compare our methods with the baselines. The experimental results are shown in Table 2. It can be seen that our proposed method outperforms the baselines in all performance measures of interest. Specifically, in the field of computer security, the recall is a very important measure of completeness since a higher recall value leads to fewer vulnerable functions being incorrectly classified as non-vulnerable, which can otherwise present an issue for code auditors when there can be a large imbalance in the number of non-vulnerable and vulnerable functions in

---

[4]https://github.com/dascimal-org/MDSeqVAE

real-world use. In addition, the fact that the resulting data representations of MDSAE-RKL and MDSAE-RWS work well with a linear classifier confirms our intuition and motivation of that the encouragement of data separation effectively supports the classifiers.

| Datasets | Windows | | | | | Linux | | | | | Whole | | | | |
|---|---|---|---|---|---|---|---|---|---|---|---|---|---|---|---|
| Methods | Acc | Rec | Pre | F1 | AUC | Acc | Rec | Pre | F1 | AUC | Acc | Rec | Pre | F1 | AUC |
| RNN-R | 54.1 | 92.6 | 52.6 | 67.0 | 53.8 | 55.3 | 93.5 | 53.3 | 67.9 | 54.9 | 56.3 | 93.9 | 53.9 | 68.5 | 55.8 |
| Para2Vec | 55.5 | 93.5 | 53.4 | 68.0 | 55.0 | 55.8 | 92.1 | 53.6 | 67.8 | 55.5 | 54.9 | 94.3 | 53.1 | 67.7 | 54.4 |
| MD-RKL | 80.8 | 86.9 | 77.6 | 82.0 | 80.7 | 82.7 | 81.3 | **83.9** | 82.6 | 82.7 | 75.3 | 87.8 | 70.5 | 78.2 | 75.1 |
| MD-RWS | 80.6 | 91.3 | 75.5 | 82.6 | 80.6 | 84.7 | 90.7 | 81.2 | 85.7 | 84.6 | 83.7 | 94.3 | 78.0 | 85.4 | 83.5 |
| RNN-C | 81.5 | 94.6 | 75.1 | 83.7 | 81.4 | 84.4 | 96.9 | 77.7 | 86.3 | 84.2 | 83.4 | 94.1 | 77.8 | 85.2 | 83.3 |
| VulDeePeck | 82.5 | 94.4 | 76.5 | 84.5 | 82.4 | 85.5 | 94.2 | 80.5 | 86.8 | 85.4 | 83.5 | 91.0 | **79.5** | 84.8 | 83.4 |
| SeqVAE-C | 80.8 | 91.4 | 75.7 | 82.8 | 80.7 | 83.0 | 93.7 | 77.5 | 84.8 | 82.9 | 78.5 | 89.4 | 73.6 | 80.7 | 78.4 |
| MD-CKL | 83.2 | **97.7** | 75.8 | 85.4 | 83.0 | 85.9 | 97.2 | 79.5 | 87.4 | 85.7 | 82.3 | 98.0 | 74.8 | 84.8 | 82.1 |
| MD-CWS | **84.5** | 97.2 | **77.7** | 86.4 | **84.4** | **86.9** | 97.8 | 80.6 | **88.3** | 86.8 | 85.3 | 98.1 | 78.4 | **87.1** | 85.2 |

Table 2: The experimental results in percent (%) of the proposed methods compared with the baselines on the NDSS18 *binary* dataset. Acc, Rec, and Pre are shorthand for the performance measures accuracy, recall, and precision, respectively.

### 4.4.2 INSPECTIONS OF MODEL BEHAVIORS

**Distances between Two Priors, Distributions of Vulnerable, Non-vulnerable Classes During Training** In this experiment, we study i) the L2 WS distance between the two priors, ii) the Euclidean distance of two means of priors (i.e., $\|\boldsymbol{\mu}_0 - \boldsymbol{\mu}_1\|$), iii) the KL divergence of $q_\phi\left(\mathbf{z} \mid \mathbf{h}_L, y = 0\right)$ and $p^0\left(\mathbf{z}\right)$ (i.e., $D_{KL}\left(q_\phi\left(\mathbf{z} \mid \mathbf{h}_L, y = 0\right) \| p^0\left(\mathbf{z}\right)\right)$), iv) the KL divergence of $q_\phi\left(\mathbf{z} \mid \mathbf{h}_L, y = 1\right)$ and $p^1\left(\mathbf{z}\right)$ (i.e., $D_{KL}\left(q_\phi\left(\mathbf{z} \mid \mathbf{h}_L, y = 1\right) \| p^1\left(\mathbf{z}\right)\right)$), v) the Maximum Mean Discrepancy (MMD) distance (Gretton et al., 2012) of $q_\phi\left(\mathbf{z} \mid \mathbf{h}_L, y = 0\right)$ and $q_\phi\left(\mathbf{z} \mid \mathbf{h}_L, y = 1\right)$, and vi) the reconstruction loss across epochs of MDSAE-RWS– the variant of our proposed method for learning separable representations. As shown in Figure 3, during the training process, two distributions $p\left(\mathbf{z} \mid y = 0\right)$ and $p\left(\mathbf{z} \mid y = 1\right)$ become consistently and gradually more distant with the increase in their MMD distance (Figure 3, second row, middle), hence implying the gradually increasing separation of the corresponding latent codes. In addition, as we expect, the two priors become consistently and gradually more distant (Figure 3, first row, left-hand side and Figure 3, first row, middle) and the latent codes of vulnerable ($y = 1$) and non-vulnerable ($y = 0$) classes become more compressed into its priors respectively (Figure 3, first row, right-hand side and Figure 3, second row, left-hand side). Furthermore, the reconstruction error consistently decreases which implies that the latent codes maintain crucial information of the original binaries (Figure 3, second row, right-hand side).

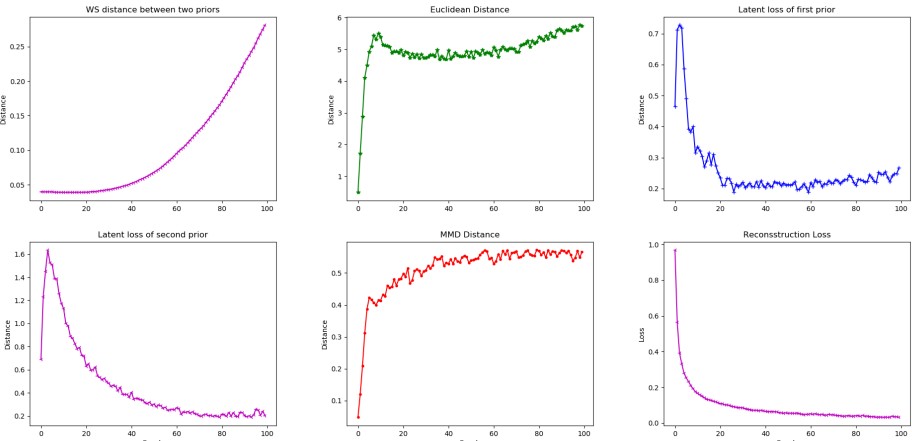

Figure 3: The L2 WS distance between two priors (first row, left-hand side), ii) the Euclidean distance of two means of priors (i.e., $\|\boldsymbol{\mu}_0 - \boldsymbol{\mu}_1\|$) (first row, middle), the KL divergence between $q_\phi\left(\mathbf{z} \mid \mathbf{h}_L, y = 0\right)$ and $p^0\left(\mathbf{z}\right)$ (i.e., $D_{KL}\left(q_\phi\left(\mathbf{z} \mid \mathbf{h}_L, y = 0\right) \| p^0\left(\mathbf{z}\right)\right)$) (first row, right-hand side), the KL divergence of $q_\phi\left(\mathbf{z} \mid \mathbf{h}_L, y = 1\right)$ and $p^1\left(\mathbf{z}\right)$ (i.e., $D_{KL}\left(q_\phi\left(\mathbf{z} \mid \mathbf{h}_L, y = 1\right) \| p^1\left(\mathbf{z}\right)\right)$) (second row, left-hand side), the MMD distance of $q_\phi\left(\mathbf{z} \mid \mathbf{h}_L, y = 0\right)$ and $q_\phi\left(\mathbf{z} \mid \mathbf{h}_L, y = 1\right)$ (second row, middle), and the reconstruction loss (second row, right-hand side) across epochs.

**Visualization of Latent Codes of Two Classes in The Latent Space** In this experiment, we set the dimension of the latent space to 2 to visualize the latent codes of the two classes before and after

training. As shown in Figure 4, the latent codes of the two classes are intermingled before training, whereas, they become more separable and distinct after training. This shows that our proposed methods discover data representations that support the classification task.

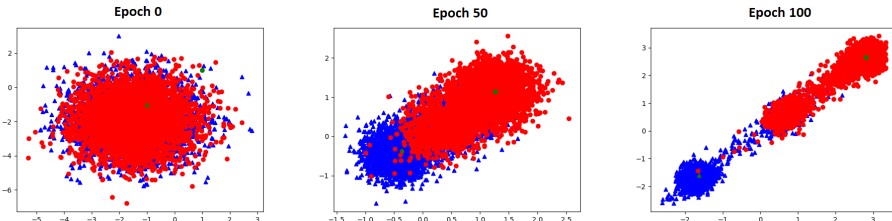

Figure 4: The 2D latent codes in the latent space before (left) and after (right) training. The green points are the means of two distributions $q_\phi \left( \mathbf{z} \mid \mathbf{h}_L, y = 0 \right)$ and $q_\phi \left( \mathbf{z} \mid \mathbf{h}_L, y = 1 \right)$.

## 5 CONCLUSION

The detection of vulnerabilities in binary code is an important problem in the software industry and in the field of computer security. In this paper, we leverage recent advances in deep learning representation to propose the Maximal Divergence Sequential Auto-Encoder for binary vulnerability detection. Specifically, latent codes representing vulnerable and non-vulnerable binaries are encouraged to be maximally different, while still being able to maintain crucial information from the original binaries. To address the issue of limited labelled public binary datasets for this problem and to facilitate research in the application of machine learning and deep learning to the domain of binary vulnerability detection, we have created a labelled binary software dataset. Furthermore, our developed tool and approach can be reused to create other high-quality binary datasets. We conducted extensive experiments to compare our proposed methods with the baselines. The experimental results show that our proposed methods outperform the baselines in all performance measures of interest.

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

## A    APPENDIX: THE PROCESS TO OBTAIN BINARIES FROM SOURCE CODE

The process of compiling the VulDeePecker dataset into binaries is divided into three main stages: collecting functions' source code, detecting and fixing source code, and compiling source code to binary functions.

### A.1    COLLECTING FUNCTIONS' SOURCE CODE

The source code is collected from VulDeePecker GitHub[5]. This source code involve two types of vulnerability in C/C++ programs: *buffer error vulnerability CWE-119* (11,427 files) and *resource management error vulnerability CWE-399* (2,088 files). Figure 8 provides an example of a source code file together with its highlighted buffer error vulnerability. We then use Joern's parser [6] to identify the start and end points of each function in order to recognize the function scope. After this step, 19,009 non-vulnerable and 12,946 vulnerable functions are detected and obtained. However, there are a considerable number of functions which are identical to each other. They are either some common functions that are widely used or some unchanged functions in different versions of a particular source code file. To address this issue, these identical functions are removed. Eventually, the numbers of distinctive non-vulnerable and vulnerable functions are 6,412 and 6,592 functions respectively (See Figure 5).

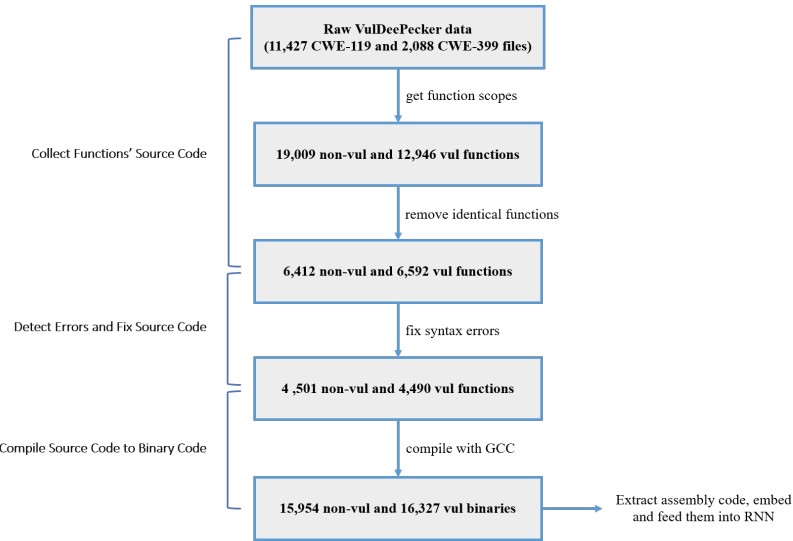

Figure 5: Detailed steps of the process of compiling VulDeePecker dataset into binaries.

### A.2    DETECTING ERRORS AND FIXING SOURCE CODE

At the second stage, as these functions are incomplete C/C++ code snippets and cannot be compiled successfully, the errors in source code are required being detected and fixed to generate binaries. Therefore, we develop an automatic tool based on Joern to detect and fix these functions. The activity diagram of our tool is described in Figure 6. In the targeted directory, our automatic tool reads every file (each contains the source code of a function) sequentially. The process of detecting and fixing each function commences with the preprocessing of source code, which adds some necessary C/C++ libraries and the main function. It is worth noting that at this step, our tool is able to reformat the source code using the clang-format [7]. Subsequently, the tool invokes the gcc/g++ (MinGW) compiler to compile the C/C++ source code respectively. The compiler then captures the error messages and calls the corresponding solver for each specific error. The semantic and syntactic relationships of the error messages with respect to the source code are mainly analyzed by Joern's parser. A function's

---

[5]https://github.com/CGCL-codes/VulDeePecker

[6]http://mlsec.org/joern/

[7]https://clang.llvm.org/docs/ClangFormat.html

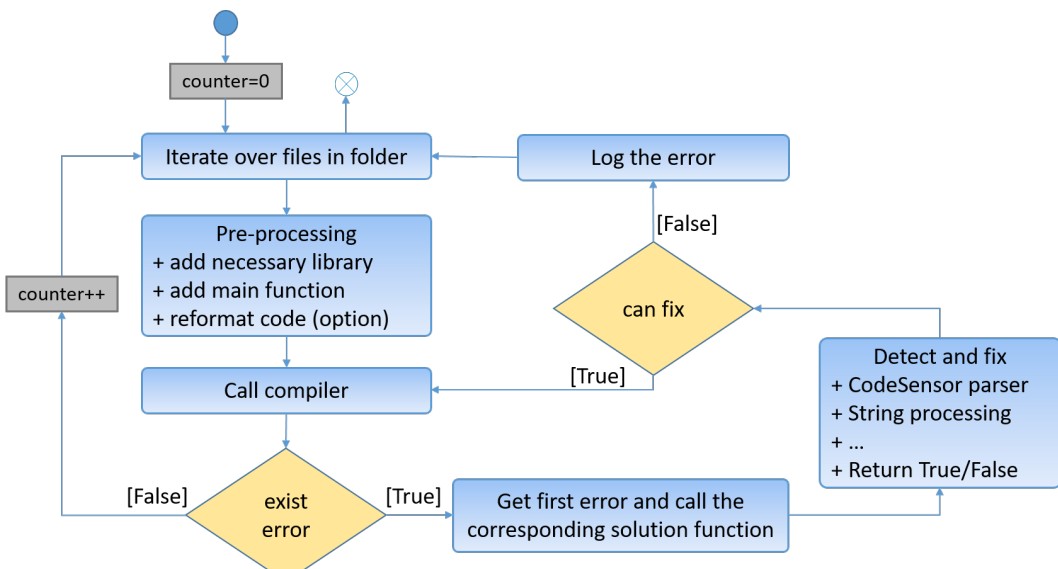

Figure 6: The activity diagram for detecting and fixing functions with syntax errors.

source code is fixed successfully and ready to compile when the compiler cannot issue any error messages in the process of detecting it.

The process of detecting errors and fixing source code also has its own challenges. As mentioned before, we collect the code snippets of the functions which are always incomplete. This leads to the missing of the declarations for some objects due to the lack of certain libraries to which those objects belong. Figure 7 refers to a typical example of an uncompilable function. For that code, when our automatic tool starts detecting the error, the gcc/g++ (MinGW) compiler informs the following error message which needs to be fixed at line 23: *'CWE761_Free_Pointer_Not_at_Start_of_Buffer__wchar_t_environment_34_unionType'* *has no member named 'unionFirst'*. The error information is then sent to Joern's parser in order to analyze and find the appropriate solution to fix this error. Unfortunately, the parser is not confident enough to declare the *'unionFirst'* variable as a member of the *'CWE761_Free_Pointer_Not_at_Start_of_Buffer__wchar_t_environment_34_unionType'* class. The reason arising from the fact that the *'myUnion.unionFirst'* was assigned to the *'data'* variable, but the parser cannot give any information about the data type of this variable. In this situation, the error description is logged into a log file, the execution of the function including this error is skipped, and our tool proceeds to the next function. After detecting and fixing errors process, we synthesize and do some statistics from the log file to know which errors account for the most popular quantities to upgrade promptly our tool. We also consider the complexity of functions and the priority order of errors to ensure errors fixed from easiest to hardest. After each upgrade, our automatic tool is very likely to detect and fix more complex errors to become more completed and stable.

The result we obtain from this stage is 8,991 fixed source code. It is noticeable that while an original function's source code has 40 errors on average, our automatic tool is able to detect and fix up to 22 and 28 general errors for each C and C++ source code respectively.

### A.3    COMPILING SOURCE CODE TO BINARY CODE

At the last stage, we compile 8,991 fixed functions which contain 4,501 non-vulnerable and 4,490 vulnerable functions into binaries under two platforms (Windows and Ubuntu) and architectures (x86/x64). The process of compilation raises a certain number of small errors due to the behavior inconsistency of gcc/g++ between two platforms which leads to the fact that some fixed functions cannot be compiled. The total number of binary functions are 32,281 wherein 15,954 functions are non-vulnerable binaries and 16,327 functions are vulnerable binaries. As the consequence, we utilize the Capstone software to disassemble these binaries into assembly code. Figure 9 shows the assembly code together with its highlighted buffer error vulnerability of the source code file

```
1    typedef struct CWE761_Free_Pointer_Not_at_Start_of_Buffer__wchar_t_environment_34_unionType{
2    }CWE761_Free_Pointer_Not_at_Start_of_Buffer__wchar_t_environment_34_unionType;
3    #include <string.h>
4    #include <stdlib.h>
5    static void goodB2G()
6    {
7        wchar_t * data;
8        CWE761_Free_Pointer_Not_at_Start_of_Buffer__wchar_t_environment_34_unionType myUnion;
9        data = (wchar_t *)malloc(100*sizeof(wchar_t));
10       data[0] = L'\0';
11       {
12           /* Append input from an environment variable to data */
13           size_t dataLen = wcslen(data);
14           int ENV_VARIABLE;
15           wchar_t * environment = GETENV(ENV_VARIABLE);
16           /* If there is data in the environment variable */
17           if (environment != NULL)
18           {
19               /* POTENTIAL FLAW: Read data from an environment variable */
20               wcsncat(data+dataLen, environment, 100-dataLen-1);
21           }
22       }
23       myUnion.unionFirst = data;
24       {
25           wchar_t * data = myUnion.unionSecond;
26           {
27               size_t i;
28               /* FIX: Use a loop variable to traverse through the string pointed to by data */
29               for (i=0; i < wcslen(data); i++)
30               {
31                   if (data[i] == SEARCH_CHAR)
32                   {
33                       printLine("We have a match!");
34                       break;
35                   }
36               }
37               free(data);
38           }
39       }
40   }
41
42   int main()
43   {
44       return 0;
45   }
```

Figure 7: Example of an uncompilable function.

in Figure 8. Overall, Figure 5 shows the details of the stages in VulDeePecker dataset processing, together with the number of vulnerable and non-vulnerable functions obtained at the end of each stage.

```
1    int SNPRINTF(){}
2    int printLine(){}
3    int staticReturnsTrue(){}
4    #include <string.h>
5    #include <stdlib.h>
6    void CWE134_Uncontrolled_Format_String__char_console_snprintf_08_bad()
7    {
8        char * data;
9        char dataBuffer[100] = "";
10       data = dataBuffer;
11       if(staticReturnsTrue())
12       {
13           {
14               /* Read input from the console */
15               size_t dataLen = strlen(data);
16               /* if there is room in data, read into it from the console */
17               if (100-dataLen > 1)
18               {
19                   /* POTENTIAL FLAW: Read data from the console */
20                   int stdin;
21                   if (fgets(data+dataLen, (int)(100-dataLen), stdin) != NULL)
22                   {
23                       /* The next few lines remove the carriage return from the string that is
24                        * inserted by fgets() */
25                       dataLen = strlen(data);
26                       if (dataLen > 0 && data[dataLen-1] == '\n')
27                       {
28                           data[dataLen-1] = '\0';
29                       }
30                   }
31                   else
32                   {
33                       printLine("fgets() failed");
34                       /* Restore NUL terminator if fgets fails */
35                       data[dataLen] = '\0';
36                   }
37               }
38           }
39       }
40       if(staticReturnsTrue())
41       {
42           {
43               char dest[100] = "";
44               /* POTENTIAL FLAW: Do not specify the format allowing a possible format string vulnerability */
45               SNPRINTF(dest, 100-1, data);
46               printLine(dest);
47           }
48       }
49   }
50
51   int main()
52   {
53       return 0;
54   }
```

Figure 8: Example of a source code file from VulDeePecker dataset together with its highlighted buffer error vulnerability.

```
0000000000000012 <CWE134_Uncontrolled_Format_String__char_console_snprintf_08_bad>:
  12:   55                        push   rbp
  13:   57                        push   rdi
  14:   48 81 ec 18 01 00 00      sub    rsp,0x118
  1b:   48 8d ac 24 80 00 00      lea    rbp,[rsp+0x80]
  22:   00
  23:   48 c7 45 10 00 00 00      mov    QWORD PTR [rbp+0x10],0x0
  2a:   00
  2b:   48 8d 55 18               lea    rdx,[rbp+0x18]
  2f:   b8 00 00 00 00            mov    eax,0x0
  34:   b9 0b 00 00 00            mov    ecx,0xb
  39:   48 89 d7                  mov    rdi,rdx
  3c:   f3 48 ab                  rep stos QWORD PTR es:[rdi],rax
  3f:   48 89 fa                  mov    rdx,rdi
  42:   89 02                     mov    DWORD PTR [rdx],eax
  44:   48 83 c2 04               add    rdx,0x4
  48:   48 8d 45 10               lea    rax,[rbp+0x10]
  4c:   48 89 85 88 00 00 00      mov    QWORD PTR [rbp+0x88],rax
  53:   e8 b4 ff ff ff            call   c <staticReturnsTrue>
  58:   85 c0                     test   eax,eax
  5a:   0f 84 d7 00 00 00         je     137 <CWE134_Uncontrolled_Format_String__char_console_snprintf_08_bad+0x125>
  60:   48 8b 85 88 00 00 00      mov    rax,QWORD PTR [rbp+0x88]
  67:   48 89 c1                  mov    rcx,rax
  6a:   e8 00 00 00 00            call   6f <CWE134_Uncontrolled_Format_String__char_console_snprintf_08_bad+0x5d>
  6f:   48 89 85 80 00 00 00      mov    QWORD PTR [rbp+0x80],rax
  76:   b8 64 00 00 00            mov    eax,0x64
  7b:   48 2b 85 80 00 00 00      sub    rax,QWORD PTR [rbp+0x80]
  82:   48 83 f8 01               cmp    rax,0x1
  86:   0f 86 ab 00 00 00         jbe    137 <CWE134_Uncontrolled_Format_String__char_console_snprintf_08_bad+0x125>
  8c:   48 8b 85 80 00 00 00      mov    rax,QWORD PTR [rbp+0x80]
  93:   ba 64 00 00 00            mov    edx,0x64
  98:   29 c2                     sub    edx,eax
  9a:   89 d0                     mov    eax,edx
  9c:   48 8b 95 80 00 00 00      mov    rdx,QWORD PTR [rbp+0x80]
  a3:   48 8b 8d 88 00 00 00      mov    rcx,QWORD PTR [rbp+0x88]
  aa:   48 01 d1                  add    rcx,rdx
  ad:   8b 55 7c                  mov    edx,DWORD PTR [rbp+0x7c]
  b0:   41 89 d0                  mov    r8d,edx
  b3:   89 c2                     mov    edx,eax
  b5:   e8 00 00 00 00            call   ba <CWE134_Uncontrolled_Format_String__char_console_snprintf_08_bad+0xa8>
  ba:   48 98                     cdqe
  bc:   48 85 c0                  test   rax,rax
  bf:   74 56                     je     117 <CWE134_Uncontrolled_Format_String__char_console_snprintf_08_bad+0x105>
  c1:   48 8b 85 88 00 00 00      mov    rax,QWORD PTR [rbp+0x88]
  c8:   48 89 c1                  mov    rcx,rax
  cb:   e8 00 00 00 00            call   d0 <CWE134_Uncontrolled_Format_String__char_console_snprintf_08_bad+0xbe>
  d0:   48 89 85 80 00 00 00      mov    QWORD PTR [rbp+0x80],rax
  d7:   48 83 bd 80 00 00 00      cmp    QWORD PTR [rbp+0x80],0x0
  de:   00
  df:   74 56                     je     137 <CWE134_Uncontrolled_Format_String__char_console_snprintf_08_bad+0x125>
  e1:   48 8b 85 80 00 00 00      mov    rax,QWORD PTR [rbp+0x80]
  e8:   48 8d 50 ff               lea    rdx,[rax-0x1]
  ec:   48 8b 85 88 00 00 00      mov    rax,QWORD PTR [rbp+0x88]
  f3:   48 01 d0                  add    rax,rdx
  f6:   0f b6 00                  movzx  eax,BYTE PTR [rax]
  f9:   3c 0a                     cmp    al,0xa
  fb:   75 3a                     jne    137 <CWE134_Uncontrolled_Format_String__char_console_snprintf_08_bad+0x125>
  fd:   48 8b 85 80 00 00 00      mov    rax,QWORD PTR [rbp+0x80]
 104:   48 8d 50 ff               lea    rdx,[rax-0x1]
 108:   48 8b 85 88 00 00 00      mov    rax,QWORD PTR [rbp+0x88]
 10f:   48 01 d0                  add    rax,rdx
 112:   c6 00 00                  mov    BYTE PTR [rax],0x0
 115:   eb 20                     jmp    137 <CWE134_Uncontrolled_Format_String__char_console_snprintf_08_bad+0x125>
 117:   48 8d 0d 00 00 00 00      lea    rcx,[rip+0x0]        # 11e <CWE134_Uncontrolled_Format_String__char_console_snprintf_08_bad+0x10c>
 11e:   e8 e3 fe ff ff            call   6 <printLine>
 123:   48 8b 85 80 00 00 00      mov    rax,QWORD PTR [rbp+0x80]
 12a:   48 8b 95 88 00 00 00      mov    rdx,QWORD PTR [rbp+0x88]
 131:   48 01 d0                  add    rax,rdx
 134:   c6 00 00                  mov    BYTE PTR [rax],0x0
 137:   e8 d0 fe ff ff            call   c <staticReturnsTrue>
 13c:   85 c0                     test   eax,eax
 13e:   74 4d                     je     18d <CWE134_Uncontrolled_Format_String__char_console_snprintf_08_bad+0x17b>
 140:   48 c7 45 a0 00 00 00      mov    QWORD PTR [rbp-0x60],0x0
 147:   00
 148:   48 8d 55 a8               lea    rdx,[rbp-0x58]
 14c:   b8 00 00 00 00            mov    eax,0x0
 151:   b9 0b 00 00 00            mov    ecx,0xb
 156:   48 89 d7                  mov    rdi,rdx
 159:   f3 48 ab                  rep stos QWORD PTR es:[rdi],rax
 15c:   48 89 fa                  mov    rdx,rdi
 15f:   89 02                     mov    DWORD PTR [rdx],eax
 161:   48 83 c2 04               add    rdx,0x4
 165:   48 8b 95 88 00 00 00      mov    rdx,QWORD PTR [rbp+0x88]
 16c:   48 8d 45 a0               lea    rax,[rbp-0x60]
 170:   49 89 d0                  mov    r8,rdx
 173:   ba 63 00 00 00            mov    edx,0x63
 178:   48 89 c1                  mov    rcx,rax
 17b:   e8 80 fe ff ff            call   0 <SNPRINTF>
 180:   48 8d 45 a0               lea    rax,[rbp-0x60]
 184:   48 89 c1                  mov    rcx,rax
 187:   e8 7a fe ff ff            call   6 <printLine>
 18c:   90                        nop
 18d:   48 81 c4 18 01 00 00      add    rsp,0x118
 194:   5f                        pop    rdi
 195:   5d                        pop    rbp
 196:   c3                        ret
```

Figure 9: The vulnerability highlighted assembly code of the corresponding function's source code in Figure 8.

