# OpenReview forum: "Maximal Divergence Sequential Autoencoder for Binary Software Vulnerability Detection"
_ICLR.cc/2019/Conference_

### Official Review · AnonReviewer2 · 2018-11-01
**Paper that addresses a new application with Deep models: binary code for vulnerability detection. One key contribution is to create a dataset for the community actually built from an existing dataset for source code vulnerability detection. A model of variational-autoencoders maximizing a divergence between positive and negative distributions is also proposed - good results on the proposed datasets are reported.**

**Rating:** 6
**Confidence:** 2

**Review:**

The paper proposes a method to classify vulnerable and non-vulnerable binary codes where each data instance is a binary code corresponding to a sequence of machine instructions. The contributions include the creation of a new dataset for binary code vulnerability detection and the proposition of an architecture based on a supervised adaptation of variational auto-encoder, built upon the result of a sequential information,
and using a regularization term to better discriminate positive from negative data. An experimental evaluation on the data proposed is presented, including several baselines, the results show the good behavior of the method.

Pros:
-Presentation of new application of representation learning models
-Construction of a new dataset to the community for binary software vulnerability detection
-The proposed model shows a good performance
Cons:
-The presentation of the dataset is for me rather limited while it is a significant contribution for the authors, it seems to be an extension of an existing dataset for source code vulnerability detection.
-From the last remark, it is unclear for me if the dataset is representative of binary code vulnerability problem
-The proposed architecture is reasonable and maybe new, but I find it natural with respect to existing work in the literature.

Comments:

-If providing a new dataset is a key contribution, the authors should spend more time to present the dataset. What makes it interesting/novel/challenging must be clarified.
This dataset seems actually built from the existing NDSS18 dataset for source code vulnerability detection. If I understood correctly, the authors have compiled (and sometimes corrected) the source to create binaries, then they use the labels in NDSS18 to label the binary codes obtained.
This a good start and can be useful for the community.
However the notion of vulnerability is not defined and it is difficult for me to evaluate the interest of the dataset.
I am not an expert in the field, but I am not that convinced that vulnerability for binary codes is necessary related to vulnerability that can be detected from source codes.
Indeed, one can think that some vulnerability may appear in binary codes that cannot be detected from source codes: e.g. use of unstable libraries, problems with specific CPU architectures, problems du to different interpretation of standard.

The current version of dataset seems to be a data where one tries to find the vulnerability that can be detected from code. It would be interesting here to know if detecting the vulnerabilities are easier from source code or from binary code.

It could be good if the authors could discuss more this point.

-The architecture proposed by the authors seems to use a sequential model (RNN or other) as indicated in Fig.2, the authors should precise this point.
The architecture is general enough to work on other problems/tasks - which is good - but the authors focus on the binary vulnerability code dataset in the experiments.

If the authors think that their contribution is to propose a general method for sequence classification, it could be good to apply it on other datasets.
Otherwise, something maybe more specific to the task would be useful.
In particular, there is no clear discussion to justify that variational autoencoders are better models for the task selected, it coud be good to argue more about it.

That being said, having non fixed priors and trying to maximize the divergence between positive and negative distributions are good ideas, but finally rather natural.

---

> ### Author Response · Authors · 2018-11-17
> **Response to AnonReviewer2**
>
> We are grateful for the reviewer’s constructive comments.
>
> * Dataset and challenging points
> - Providing a new dataset for binary software vulnerability detection (SVD) is one of the key contributions of this paper. Deep learning has enjoyed great success in the domains of computer vision, speech recognition, and natural language processing. However, deep learning has only had limited applications in the cyber security domain, especially in the case of binary SVD — an important and difficult problem in cybersecurity. The reason is due to the scarcity of qualified datasets for source and binary code. For collecting vulnerable source code, one needs to access the CVE website and then navigate to the relevant websites to manually collect the source code. Due to the nature of data collection, the various source codes in vulnerability detection are collected from many different software libraries, packages are always incomplete with many missing variables, data types, functions, class declarations and so on. The challenge here is how to parse the source code to determine its structure given the fact that the source code has yet to be compiled. Note that for compilable source code, one can use the ROSE parser to completely parse the code ’s structure.
> - To fix any source code, we first compiled the code using the gcc/g++ (MinGW) compiler and based on the error messages to gradually fix the code. For example, when the compiler gave an error message for a missing declaration of an identifier, our tool needs to parse the source code to understand the type of this identifier (e.g., a variable, data type, function, class) and then provide the declaration for this identifier. Still, the prototypes of data types, functions, and classes might be various enormously and be complex. Our automatic tool tackles these issues. We provide more clarification and explanation of how our tool works in the appendix section of the revised version.
>
> * Necessity of SVD for binary codes
> - In practice, binary vulnerability detection (VD) is more relevant and impactful than source code vulnerability detection. The reason is that when using a commercial application, we generally only possess its binary code and usually not its source code.  Because of license copyrights, the binary code cannot be decompiled back to the source code for VD. We, therefore, need to detect vulnerabilities at the binary level.
>
> * SVD is easier from source code or from binary code?
> - Compared with source code VD, binary code VD is significantly more difficult because much of the syntax and semantic information provided by high-level programming languages is lost during the compilation process. For source code, one can take advantage of the syntax and semantic information to construct tree structures (e.g. abstract syntax tree). In contrast, a binary code has much less syntax and semantic information as they can be only viewed as a sequence of machine instructions or bytes.
>
> * The architecture in Fig.2
> - In Fig. 2, we used a simple RNN for demonstration purposes. However, we could use a Bidirectional RNN with GRU or LSTM cells. In our experiments, we employed a dynamic RNN with GRU cell.
>
> * Apply proposed method on other datasets or something maybe more specific to the task would be useful
> - Our proposed method is general enough for application to other popular problems like sentiment analysis. We proposed a specific embedding technique for binary SVD (see Sec. 3.1) where the format of machine instructions was considered.
> - In this work, we want to demonstrate the applicability of deep learning (or machine learning) to SVD — which is an important and complex problem in cybersecurity. With the dataset provision and preliminary experimental results, we hope to further encourage the application of deep learning (machine learning) to SVD and other cybersecurity applications.
>
> * Variational autoencoders are better models for the task
> - The two key aspects that contribute to the success of our proposed model are i) the capability to reconstruct sequential binary codes from their latent representations via the VAE formulation and ii) the ability to maximize the divergence of latent representations by “pushing away” the two learnable priors. The first aspect ensures that the latent representations can capture the crucial information in the original binary codes. We observe that a vulnerable binary and its fixed version only differ by a few machine instructions, hence the ability to be able to reconstruct is important to differentiate a vulnerable binary and its fixed version in the latent space because the model needs to pay attention to the slight difference in vulnerable and fixed binaries to successfully reconstruct them. In addition, by maximizing the divergence between two learnable priors, the latent representations of vulnerable and non-vulnerable binaries are encouraged to be maximally divergent for classification purpose.

---

### Official Review · AnonReviewer3 · 2018-11-06
**This paper proposes a model to automatically extracted features for vulnerability detection using deep learning technique.**

**Rating:** 6
**Confidence:** 3

**Review:**

This paper proposes a model to automatically extracted features for vulnerability detection using deep learning technique.

Pros:
+ Create a labeled dataset for binary code vulnerability detection and attempts to solve the difficult but practical task of vulnerability detection.
+ Expend VAE from single prior to multiple priors.
+ Using figures and visualizations to show the behaviors of model.

Cons:
- The operation that creates dataset may introduce bias or variance. (The developed tool that automatically detects the syntactical errors in a given piece of source code, fixes them, and finally compiles the fixed source code into binaries, may change the distribution of data.) Why not follow the way of producing dataset of malware detection or other tasks that using binary code.
- It seems that the proposed model fails to consider the properties of the binary codes in this task. It would be more interesting if some design incorporates the special properties of the task.
- The discussion in Figure 2 and equations  are unclear. More explanations are needed. e.g. how to testing with label-unknown data.
- Many typos are found. E.g., : the given given  -> the given;    k = 1,2 should be k = 0,1

---

> ### Author Response · Authors · 2018-11-17
> **Response to AnonReviewer3**
>
> We are grateful for the reviewer’s constructive comments.
>
> * The operation that creates dataset may introduce bias or variance. (The developed tool that automatically detects the syntactical errors in a given piece of source code, fixes them, and finally compiles the fixed source code into binaries, may change the distribution of data.) Why not follow the way of producing a dataset of malware detection or other tasks that using binary code.
> - The task of creating a binary dataset for vulnerability detection is much harder and complex compared to that of malware detection. For malware detection, one can easily collect infectious binaries, while vulnerable software code requires domain experts with relevant expertise to inspect the software and labeling them (identify the location(s) of the vulnerability, the vulnerability type etc.). Since binary codes are less informative and even experts are generally unable to label them directly, only vulnerable source codes are available. However, the process to collect vulnerable source code is labor-intensive and manual since one needs to go to the CVE website (https://cve.mitre.org/cgi-bin/cvename.cgi?name=CVE-2018-9989) and then navigate to relevant websites to manually collect the source code. Due to the nature of data collection, the various source codes in vulnerability detection are collected from many different software libraries, packages are always incomplete with many missing variables, data types, functions, class declarations and so on. Each missing code item might itself have large variations in its prototype and signature. Our tool needs to parse the source codes and be aware of the relationship between the missing code item/chunk and its role in order to fix the source code given the fact that the source codes have not compiled yet. It is worth noting that for compilable source code, one can use the ROSE parser (https://en.wikipedia.org/wiki/ROSE_(compiler_framework) ) to completely parse their structures.
> - To fix any source code, we first compiled the code using the gcc/g++ (MinGW) compiler and based on the error messages to gradually fix the code. For example, when the compiler gave an error message for a missing declaration of an identifier, our tool needs to parse the source code to understand the type of this identifier (e.g., a variable, data type, function, class) and then provide the declaration for this identifier. Still, the prototypes of data types, functions, and classes might be various enormously and be complex. Our automatic tool tackles these issues.
> - Our tool does not change the execution flow, semantic and syntactic structure of the source code which are crucial in identifying vulnerabilities. Thus, the fixing and compiling process does not affect vulnerabilities inside the source codes.
>
> * It seems that the proposed model fails to consider the properties of the binary codes in this task. It would be more interesting if some design incorporates the special properties of the task.
> - Source code has useful information with the syntax and semantic information provided by high-level programming languages and one can take advantage of the syntax and semantic information to construct tree structures like, for example, the abstract syntax tree. In contrast, a binary code has reduced syntax and semantic information wherein they can be only viewed as a sequence of machine instructions or bytes.
> - In our proposed method, the embedding component (see Section 3.1) of the machine instruction format was designed to capture the properties of the binary code. We observed a significant improvement in predictive performance when using this embedding technique instead of using a standard word embedding.
>
> * The discussion in Figure 2 and equations are unclear. More explanations are needed. e.g. how to testing with label-unknown data.
> - The key aspects of our method are i) how to develop the sequential VAE to encode binaries and ii) how to maximize the latent codes of the different classes by “pushing away” the learnable priors. For testing a binary with an unknown label, we first feed it into the RNN encoder to work out its latent code and then use the classifier acting on the latent space to classify the binary. We will revise this technical section to clarify.
>
> * Many typos are found. E.g., : the given given  -> the given;    k = 1,2 should be k = 0,1
> - Thanks. We will fix them in the revised version.

---

### Official Review · AnonReviewer1 · 2018-11-06
**Potentially useful dataset of compiled code snippets classified as "vulnerable" or "not vulnerable", interesting story emerges from evaluation of family of classifiers evaluated on this dataset**

**Rating:** 7
**Confidence:** 2

**Review:**

This paper sets out to classify source code snippets are “vulnerable” or “not vulnerable” using sequential auto-encoders with two latent distributions (corresponding to the output classes), regularized to maximize divergence between theses two distributions (named Maximal Divergence Sequential Auto-Encoder).  The authors created a compiled subset of the NDSS18 vulnerable vs. non-vulnerable software dataset (which is listed as one of their primary contributions). The dataset construction required non-trivial effort since example code snippets are often incomplete and the authors needed to “fix” these code examples in order to compile them. The fixed code examples are then compiled against both Windows and Linux and both the x86 and x86-64 architectures. The inputs to all predictive models are the opcode sequence of the compiled programs.

This paper compares against one previously published vulnerability detection method (VulDeePecker) which is a bidirectional RNN followed by a linear classifier. They also compare with a cascade of models with increasingly complex components:

* RNN-R: A recurrent neural network trained in an unsupervised fashion (language modeling over opcode sequences), whose representations are then fed into an independent linear model.
* RNN-C: End-to-end training of a recurrent model over opcodes, followed by a single dense layer.
* Para2Vec: Encoding of the opcode sequence using the paragraph-to-vector architecture — I’m curious what they used as the paragraph boundaries in the compiled programs and whether the subsequent classifier was the same as RNN-C.
* SeqVAE-C: Sequential variational auto encoder trained end-to-end with a final classification layer.
* MDSAE-RKL:  Maximal divergence sequential auto-encoder with KL divergence between the two class’s latent distributions, final classifier trained independently.
* MDSAE-RWS: Maximal divergence sequential auto-encoder with L2/Wasserstein  divergence between the two class’s latent distributions, final classifier trained independently.
* MDSAE-CKL: Maximal divergence sequential auto-encoder with KL divergence between the two class’s latent distributions, final classifier included as the final layer of the whole model.
* MDSAE-CWS: Maximal divergence sequential auto-encoder with L2/Wasserstein  divergence between the two class’s latent distributions, final classifier included as the final layer of the whole model.

The two MDSAE models using Wasserstein divergence vastly outperform the two equivalent models using KL divergence. Another generalization that can be drawn from the evaluation is that models which are trained  with supervision end-to-end outperform those which train representation and classifier separately.

Overall, I think this is an interesting and cool paper but I’m not sure I actually buy into the basic premise that it makes sense to model vulnerable vs. non-vulnerable code as two different latent spaces. Aren’t the changes to make a vulnerable function safe again rather small and/or subtle? I think that beyond visualizing the convergence of properties of the latent spaces it would greatly improve this paper to inspect which aspects of the source contribute to both the latent representation and final classification as vulnerable vs. non-vulnerable.

Also, I wish the process of “fix”ing the input code was better described, since the failure of this procedure excluded 4k/13k of the programs/functions in their initial dataset and had the potential to introduces learnable biases in the source code. At the very least, the authors should list how many vulnerable vs. non-vulnerable samples required fixing vs. could be compiled in their original form.

Lastly, the definition of "vulnerable" may be obvious to someone more familiar with the domain but seemed to me somewhat vague and never directly addressed.

Typo:
p3, need space in "obtain32, 281"

---

> ### Author Response · Authors · 2018-11-17
> **Response to AnonReviewer 1**
>
> We are grateful for the reviewer’s constructive comments.
>
> *  I’m not sure I actually buy into the basic premise that it makes sense to model vulnerable vs. non-vulnerable code as two different latent spaces. Aren’t the changes to make a vulnerable function safe again rather small and/or subtle? I think that beyond visualizing the convergence of properties of the latent spaces it would greatly improve this paper to inspect which aspects of the source
> - The two key aspects that contribute to the success of our proposed model are i) the capability to reconstruct sequential binary codes from their latent representations via the VAE formulation and ii) the ability to maximize the divergence of latent representations by “pushing away” the two learnable priors. The first aspect ensures that the latent representations are able to capture the crucial information in the original binary codes. We observe that a vulnerable binary and its fixed version only differ by a few machine instructions, hence the ability to reconstruct is important to differentiate a vulnerable binary and its fixed version in the latent space because the model needs to pay attention to the slight difference in vulnerable and fixed binaries to successfully reconstruct them. In addition, by maximizing the divergence between two learnable priors, the latent representations of vulnerable and non-vulnerable binaries are encouraged to be maximally divergent for classification purpose.
> - We have recently found a tool referenced in a paper (http://proceedings.mlr.press/v80/chen18j.html) that could be useful for inspecting which aspects of the source and binary contribute to both the latent representation and final classification as vulnerable vs. non-vulnerable.
>
> * I wish the process of “fix”ing the input code was better described. The authors should list how many vulnerable vs. non-vulnerable samples required fixing vs. could be compiled in their original form.
> - In the appendix section, we will add the technical steps, challenges, and details of the process to compile source to binary codes. We will also list the number of vulnerable and non-vulnerable source codes required fixing and those being able to be compiled using our tool in the paper.
>
> * The definition of "vulnerable" may be obvious to someone more familiar with the domain but seemed to me somewhat vague and never directly addressed”.
> - We indeed stated the definition of a software vulnerability in the first sentence of the paper. To further clarify this, we will add some typical examples of vulnerable source and binary codes in the appendix.

---

> > ### Comment · AnonReviewer1 · 2018-11-22
> > **Appendix will be very welcome**
> >
> > An appendix with with (1) a detailed explanation of the compilation process and (2) multiple code examples (vulnerable, non-vulnerable, compiles successfully, cannot be compiled) will significantly improve this paper. I think, where possible, try to integrate this information (even at a summarized high level) into the text of the paper itself, since it will remove some of the mystery I experienced reading it.

---

> > > ### Author Response · Authors · 2018-11-22
> > > **Regarding the appendix**
> > >
> > > Many thanks for your response and comment. We are doing the proofreading for the appendix and going to submit the revised version as soon as possible.

---

### Official Review · AnonReviewer4 · 2018-11-13
**Using deep neural networks to learn embeddings for binary software vulnerability detection; unclear about its contribution for ML community**

**Rating:** 6
**Confidence:** 4

**Review:**

This paper proposes a variational autoencoder-based architecture for binary code embedding. For evaluation, they construct a dataset by compiling source code in the NDSS18 dataset. They evaluate their approaches against several neural network baselines, and demonstrate that their learned embeddings are more effective at distinguishing between vulnerable and non-vulnerable binary code.

The application of deep representation learning for (binary) vulnerability detection is  a promising direction in general. Meanwhile, the authors did a quite comprehensive comparison with neural network baselines for embedding representation. However, I have several questions and concerns about the paper:

- The contributions of this paper are unclear to me. The authors claim that a main contribution is their dataset. I agree that this is a contribution, but since this dataset is built upon an existing dataset with source code, and the dataset construction techniques themselves are not novel, especially for machine learning community, I do not see a significant contribution in this part.

- The proposed approach is new, but the technical novelty is marginal. I think this model design is not specific to the binary vulnerability detection, but should also be applicable to other vulnerability detection settings, e.g., the original NDSS18 dataset. It would be great if the proposed approach also performs better on other vulnerability detection tasks than the baselines.

- What would be the performance of using hand-designed features on the same benchmark? If the proposed approach learns better embeddings, any intuition on what additional information is captured by the learned embeddings?

Minor suggestions: The paper needs an editing pass to fix some typos. Also, the authors seem to setup the paper template in a wrong way, and may need to consider fixing it.

---

> ### Author Response · Authors · 2018-11-17
> **Response to AnonReviewer4**
>
> We are grateful for the reviewer’s constructive comments.
>
> * The contributions are unclear to me. The authors claim that the main contribution is their dataset. I agree that this is a contribution, but since this dataset is built upon an existing dataset with source code, and the dataset construction techniques themselves are not novel, especially for machine learning community
> - Deep learning has enjoyed great success in the domains of computer vision, speech recognition, and natural language processing. However, deep learning has only had limited applications in the cybersecurity domain, especially in the case of binary software vulnerability detection — an important and difficult problem in cybersecurity. The main reason is due to the lack of qualified binary datasets for the vulnerability detection task. Our contribution is to create a qualified binary dataset for this task and propose a new method that results in good performance.
> - It is arguable that the dataset is built upon an existing dataset with source code, hence the task of creating the binary dataset is perceived to be trivial. However, the various source codes in vulnerability detection are collected from many different software libraries, packages are always incomplete with many missing variables, data types, functions, class declarations and so on. Each missing code item might itself have large variations in its prototype and signature. Our tool needs to parse the source codes and be aware of the relationship between the missing code item/chunk and its role in order to fix the source code. Overall, this task can be quite complicated.
> - We believe that the two contributions alone justify the paper. The contributions are complementary, additive and they promote the study of deep learning (machine learning) for binary software vulnerability detection — an application of cybersecurity that is significantly challenging wherein deep learning (machine learning) has been restrictively applied. However, we have shown that deep learning can be successfully applied to this problem.
>
> * It would be great if the proposed approach also performs better on other vulnerability detection tasks than the baselines
> - In practice, binary vulnerability detection is more relevant and impactful than source code vulnerability detection. The reason is that when using a commercial application, we generally only possess its binary code and usually not its source code.  Because of license copyrights, the binary code cannot be decompiled back to the source code for vulnerability detection. We, therefore, need to detect vulnerabilities at the binary level. Our proposed method can also be adapted to source code vulnerability detection and we believe that the proposed method should perform better than the baselines in this application. In this paper, we mainly focus on binary vulnerability detection, which is harder and more applicable than source code vulnerability detection. As an aside, source code vulnerability detection should also have a better performance than the equivalent binary code vulnerability detection due to the fact that a lot of information (e.g., semantics) is stripped from the source code during compilation.
>
> * The performance of using hand-designed features on the same benchmark? If the proposed approach learns better embeddings, any intuition on what additional information is captured by the learned embeddings?
> - Grieco et al. 2016 proposed using dynamic features extracted from the execution of binaries and static features extracted from the binary programs and then trained a feed-forward neural network for classification. They undertook experiments using their own dataset VDiscovery with 1039 binaries. They claimed in that paper that they had to collect their own datasets because they found no suitable datasets to perform the evaluation of their technique. We cannot compare with this method because its code is not available.
> - The two key aspects that contribute to the success of our proposed model are i) the capability to reconstruct sequential binary codes from their latent representations via the VAE formulation and ii) the ability to maximize the divergence of latent representations by “pushing away” the two learnable priors. The first aspect ensures that the latent representations are able to capture the crucial information in the original binary codes. We observe that a vulnerable binary and its fixed version only differ by a few machine instructions, hence the ability to be able to reconstruct is important to differentiate a vulnerable binary and its fixed version in the latent space because the model needs to pay attention to the slight difference in vulnerable and fixed binaries to successfully reconstruct them. In addition, by maximizing the divergence between two learnable priors, the latent representations of vulnerable and non-vulnerable binaries are encouraged to be maximally divergent for classification purpose.

---

> > ### Comment · AnonReviewer4 · 2018-12-01
> > **Thank you for the clarification and revision**
> >
> > Based on the response and the revision, I would like to increase my score to 6.

---

> > > ### Author Response · Authors · 2018-12-02
> > > **Thanks for your decision**
> > >
> > > Many thanks for your decision. We appreciate this.

---

### Meta-Review · Area_Chair1 · 2018-12-15
**applied paper in interesting domain**

**Confidence:** 4
**Recommendation:** Accept (Poster)

**Metareview:**


* Strengths

This paper applies deep learning to the domain of cybersecurity, which is non-traditional relative to more common domains such as vision and speech. I see this as a strength. Additionally, the paper curates a dataset that may be of broader interest.

* Weaknesses

While the empirical results are good, there appears to be limited conceptual novelty. However, this is fine for a paper that is providing a new task in an interesting application domain.

* Discussion

Some reviewers were concerned about whether the dataset is a substantial contribution, as it is created based on existing publicly available data. However, these concerns were addressed by the author responses and all reviewers now agree with accepting the paper.